# Eco-Friendly Poly (Butylene Adipate-*co*-Terephthalate) Coated Bi-Layered Films: An Approach to Enhance Mechanical and Barrier Properties

**DOI:** 10.3390/polym16091283

**Published:** 2024-05-03

**Authors:** Raja Venkatesan, Krishnapandi Alagumalai, Alexandre A. Vetcher, Bandar Ali Al-Asbahi, Seong-Cheol Kim

**Affiliations:** 1School of Chemical Engineering, Yeungnam University, 280 Daehak-Ro, Gyeongsan 38541, Republic of Korea; miatdhu1222@gmail.com; 2Institute of Biochemical Technology and Nanotechnology, Peoples’ Friendship University of Russia n.a. P. Lumumba (RUDN), 6 Miklukho-Maklaya St., 117198 Moscow, Russia; avetcher@gmail.com; 3Department of Physics and Astronomy, College of Science, King Saud University, P.O. Box 2455, Riyadh 11451, Saudi Arabia; balasbahi@ksu.edu.sa

**Keywords:** coatings, paper, poly(butylene adipate-co-terephthalate)-(PBAT), tensile strength, food packaging

## Abstract

In this research work, a coated paper was prepared with poly (butylene adipate-co-terephthalate) (PBAT) film to explore its use in eco-friendly food packaging. The paper was coated with PBAT film for packaging using hot pressing, a production method currently employed in the packaging industry. The coated papers were evaluated for their structural, mechanical, thermal, and barrier properties. The structural morphology and chemical analysis of the coated paper confirmed the consistent formation of PBAT bi-layered on paper surfaces. Surface coating with PBAT film increased the water resistance of the paper samples, as demonstrated by tests of barrier characteristics, including the water vapor transmission rate (WVTR), oxygen transmission rate (OTR), and water contact angle (WCA) of water drops. The transmission rate of the clean paper was 2010.40 cc m^−2^ per 24 h for OTR and 110.24 g m^−2^ per 24 h for WVTR. If the PBAT-film was coated, the value decreased to 91.79 g m^−2^ per 24 h and 992.86 cc m^−2^ per 24 h. The hydrophobic nature of PBAT, confirmed by WCA measurements, contributed to the enhanced water resistance of PBAT-coated paper. This result presents an improved PBAT-coated paper material, eliminating the need for adhesives and allowing for the fabrication of bi-layered packaging.

## 1. Introduction

Paper material is extensively utilized as packaging for a variety of products, including food items, electronics, clothing, and medicines. Packaging and protecting goods for shipment utilize paper [1]. However, paper materials cannot be used as widely as they could be due to their high levels of hydrophilicity and porosity, which decrease their barrier characteristics (moisture, gases, and lipids) [2,3]. Most studies have focused on the importance of replacement packaging fabricated from petroleum-based materials [4,5,6]. The implementation of well-established manufacturing procedures that can be easily implanted is a requirement for future possibilities. The most appropriate alternatives for packaging should be biodegradable, exhibit nontoxicity, be cheap, exhibit good barrier characteristics, and drastically decrease the amount of waste plastics [7,8].

The advances in biobased materials as a suitable alternative to petroleum-based plastics have received special interest from researchers [9,10,11]. The direct use of paper in packaging is, however, limited by the following characteristics: (i) its hydrophilicity, which results in limited water resistance; (ii) low oil barrier characteristics; and (iii) environment-friendly material for microorganisms, which increases the possibility of their development if paper arrives into direct contact with food items [12]. Biodegradable materials have been used as an alternative material due to their sustainability, degradability, and environmental friendliness [13,14,15,16,17]. Paper coated with polymers is a successful method to provide moisture and gas-tightness protection in food packaging. However, it impacts both the polymer and the paper [1]. While biodegradable polymers are useful materials for gas barriers, their hydrophobicity prohibits them from being easily utilized in barrier coatings [2,18].

PBAT may have certain performance advantages that make it suitable for applications with paper coatings. This includes its flexibility, adherence to paper, barrier characteristics, and compatibility with other coatings. PBAT is produced by a polycondensation reaction using 1,4-butanediol, terephthalic acid, and adipic acid as raw materials and using organic compounds as a catalyst. Its remarkable strength and flexibility are attained via its molecular structure, which consists of both aliphatic and aromatic segments. The result allows it to become blown into a film for packaging. Rajendran and Han [19] and Naser et al. [20] believe that this polymer material can efficiently replace polyethylene. According to Jang et al. [21], the development of the food delivery and distribution sector has significantly raised the utilization of plastic in packaging over the last few decades in South Korea. On the other hand, PBAT has limited applications, high production costs, and a low crystallization rate [22], which could have been considered additional factors in choosing PBAT for paper coating. There are some things to reflect on, including barrier characteristics [3], regulatory compliance, biodegradability [23,24], suitability with paper recycling, market demand, and perception [25,26,27,28]. PBAT is a biodegradable plastic with good hydrophobic processing characteristics and notable flexibility. The tensile strength of PBAT materials has been increased with its application. It is of the utmost importance to use the appropriate materials and processing aids while modifying a PBAT in an attempt to enhance its overall performance and reduce its cost of usage [24,29].

This can be prevented by coating the paper with PBAT film, which can extend the shelf-life of food and serve as a strong barrier against pollutants, water, and oils [30,31]. In addition, it is advantageous to have an oxygen and water vapor barrier for paper, as the fibers on paper produced from recycled materials have a lower quality than virgin fibers (due to the drying process of the recycled paper leads to fiber size and pore volume of the fiber walls to drop) [32,33,34]. It has been noted that the PBAT and fiber contents of paper offer a wide range of uses. With the use of PBAT film, this paper aims to package materials by hot-pressing them onto coated paper. Two layers of PBAT film with 100 µm thickness are placed together with paper and heated to 150 °C for 30 min. The materials have good mechanical and barrier properties, rendering it an enhanced material compared to uncoated paper, and it could act as a more secure material for food packaging. A study was conducted to analyze the structure, mechanical properties, oxygen and water vapor permeability, contact angle, and food quality of coated and uncoated paper materials. The objective was to produce and evaluate PBAT film-coated paper material.

## 2. Materials and Methods

### 2.1. Materials

The molecular weight of Mw = 14.2 × 10^4^ g/mol and the melt flow index (MFI) of 3.3–6.6 g/10 min (at 190 °C; 2.16 kg) were supplied for the PBAT by M/s BASF Ltd., Tokyo, Japan. The paper with a GSM of 180 g/m^2^ and a thickness of 160 μm was supplied by Hansol Paper Ltd., Seoul, Republic of Korea. The solvents for acetone, chloroform, and glycerol were provided by Daejung Chemicals in Busan, Republic of Korea. All of the chemicals were purified before use.

### 2.2. Fabrication of PBAT Film-Coated Paper

Prior to use, the PBAT pellets had been dried in an oven at 60 °C for 24 h. Solution mixing and drop-casting were used in the fabrication of the PBAT [35,36,37,38,39]. In chloroform, 2.0 g of PBAT was dissolved. The stirred solutions were then transferred onto a glass plate after 12 h and sonicated for 30 min to form a PBAT film. The solvent was permitted to air out for a further 48 h at room temperature so as to measure its characteristics. The process to fabricate coated paper utilizing PBAT film is shown in Figure 1. To produce the paper coating, two 2 × 2 cm^2^ PBAT film were coated on the top and bottom of the paper. After that, the sample was heated to 150 °C for 15 min in a laboratory hydraulic heating press (UL Chemical, Seoul, Republic of Korea; Model ULC-HP 400M).

### 2.3. Characterization

#### 2.3.1. Thickness Measurements

The average thickness was measured with a digital Mitutoyo Absolute dial indicator (Mitutoyo, Japan) as the mean of five random measures obtained at the sample’s location.

#### 2.3.2. Structure and Morphological Studies

FTIR (Perkin-Elmer Spectrum Two) spectra in the 4000–400 cm^−1^ spectral range were used for the ATR-FTIR spectra. X-ray diffraction (Rigaku, Cedar Park, TX, USA, PANALYTICAL) was performed in the 10° to 80° 2θ scan range at a scan rate of 0.50 min^−1^. SEM (Hitachi S-4800, Tokyo, Japan) was used to study the structure of the PBAT film-coated paper. The coated paper morphology was measured with an SEM working at 15 kV.

#### 2.3.3. Thermal Characterization

A thermogravimetric analyzer (SDT Q600 of TA Instruments, Eden Prairie, MN, USA) was used to test thermal stability. In the case of TGA experiments, the samples were heated in an N_2_ atmosphere at a rate of 10 °C/min up to 700 °C. In the case of DSC experiments, the samples were preheated to 180 °C in an N_2_ atmosphere, kept for 2 min, and then cooled. Further, the samples were heated in N_2_ flow with a rate of 20 °C/min up to 300 °C.

#### 2.3.4. Mechanical Properties

In this research, the mechanical properties of coated and uncoated paper were tested. The mechanical characteristics evaluated with the TAPPI T494 [40] procedure were tensile strength and elongation at break. The specimens for the TAPPI T494 test with sizes of 25 × 180 mm^2^ were prepared by cutting both coated and uncoated paper. The studies were performed using the 3345 (Instron, Norwood, MA, USA), a universal testing instrument. It had a starting distance of 120 mm, a test speed of 20 mm/min, and a 1.0 kN load cell. The most common measurement for resistant breaking is the burst strength, also known as the pop strength. The Mullen Burst strength tester (HT 8020 A) served to measure it. To cut specimens into 100 × 100 mm^2^ sizes for the TAPPI T403 test [41], a special cutter was used. Bursting strength, measured in kPa, represents the highest strength value achieved before rupture. In the effort to calculate the burst index (KPa m^2^/g), the burst strength was divided by the grammage.

#### 2.3.5. Porosity and Water Absorption

The Cobb-60 method was used to measure the water absorption capacity of uncoated and coated paper, which was expressed in gH_2_O per m^2^ of sample. These measurements were carried out using a COBB tester (Test Techno, Kolkata, India) at room temperature via the TAPPI T456 standard procedure [42]. Samples with sizes of 10 × 10 cm were conditioned at 23 ± 1 °C and 50 ± 1% relative humidity. Dividing the Cobb value by grammage yielded the Cobb index.

The porosity of the samples was measured using a Frank PTI porosimeter via the TAPPI T 460 standard method. For this purpose, samples with a diameter of 100 mm were prepared. The porosity test of dry samples was carried out at 23 °C under a pressure of 1.47 kPa.

#### 2.3.6. Barrier Properties

An oxygen permeability tester (NOSELAB ATS, Nova Milanese, Italy) was used to determine the oxygen transmission rate (OTR) of the paper samples at 25 °C with the ASTM D3985 [43] standard procedure. After three evaluations were taken on paper at different places, the average value was calculated. For all of the samples, room temperature was maintained. MOCON’s PERMATRAN, Minneapolis, MN, USA, and the ASTM F1249 [44] method were used to measure the water vapor transmission rate (WVTR) of the paper specimens. The measurements were performed several times to calculate an average result.

#### 2.3.7. Water Contact Angle (WCA) Measurements

OCA-20 of Dataphysics Instruments was used to determine the contact angles of uncoated and coated paper samples. Water drops of 1 μL volume were used for these measurements. The images of the drops were captured for 5 s.

#### 2.3.8. Statistical Analysis

The results received were analyzed using the variation ANOVA method with the SPSS statistical software (Origin 9.0) package (SPSS Thailand Ltd., Bangkok, Thailand). The value of *p* was less than 0.05. Tukey’s multiple evaluation test was used to identify significant changes.

## 3. Results and Discussion

After drying, the PBAT-film-coated paper was uniform, and the pieces had not deformed due to the coating. The coating’s uniformity was confirmed after PBAT films were applied to the paper surfaces [45].

The fibrous structure of the paper could be packed with the coating’s PBAT film if coated to a surface [46]. The results showed that the polymer coating was able to enter the fibers, which improved the barrier properties of the coated paper. It is conceivable that the coating with PBAT films contributed to superior uniformity results compared to those described by Hashmi et al. [47]. The recyclable material with good characteristics was produced by coating a PBAT film on paper by hot-pressing it. This method is easy from an economic point of view, and rolling pressing allows the process to be carried out cost-effectively. However, chloroform is a hazardous solvent, and PBAT is difficult to dissolve; therefore, using a PBAT solution for film forming is not considered cost-effective. The properties of the paper, whether uncoated or coated with PBAT, are shown in Table 1 with the aim of measuring the properties.

### 3.1. Structural Analysis

Figure 2A represents the FTIR spectra of uncoated and coated paper samples. On the cellulose of the paper, characteristic bands linked to the -OH stretching vibration, C-O vibration, and -OH vibration occurred near 3424 cm^−1^, 1092 cm^−1^, and 1163 cm^−1^, respectively. In addition, at 1103 cm^−1^ and 935 cm^−1^, unique bands related to the C–O vibration bands were observed. These absorption bands show that the coated materials contained PBAT and paper. An increase in -OH and hydrogen bond counts could help to clarify it. The interaction between PBAT and paper can be seen through a change in absorption bands.

### 3.2. Morphological Analysis

SEM images can be used to demonstrate the effects of PBAT coatings on the paper’s morphology. Figure 3 represents the SEM images of the paper, as well as PBAT film-coated paper surfaces and cross-sections. The uncoated paper showed a rough, porous surface with voids within the cellulose fibers [50,51]. To produce an even and smooth surface, the PBAT film was bonded to the fibrous structure of the paper. Figure 3 shows that the thickness of the uncoated paper substrate was 160 ± 2.0 μm, while the thickness of the PBAT-film coated paper substrate was 439.4 ± 6.0 μm when PLA and polystyrene were coated with paper and rice straw pulp paper. Accordingly, similar results with a smooth surface were obtained [52,53]. An uneven, fibrous structure was observed in the cross-sectional images of the uncoated papers. However, there was no vacant space on the surface of the coated paper because the pores of the cellulose fibers were filled with PBAT. If the paper was coated and the surface filled with PBAT, a smooth and uniform surface was observed, providing an excellent degree of coverage over the paper with no visible pores, indicating that the polysaccharides with the rubber coating layer were compatible with the paper and adhered easily to the paper surface [23,54]. These results justify the OTR and WVTR properties of this packaging material.

### 3.3. Thermal Analysis

Thermogravimetric analysis (TGA) and differential scanning calorimetry (DSC) analysis were used to evaluate the thermal characterization of paper and PBAT film-coated paper material. These were used to analyze the thermal limits for convenient food packaging fabricated from paper coated with PBAT. The influence of the PBAT layer on the thermal stability of the coated paper material was estimated by TGA, as illustrated in Figure 4A. The weight loss of paper with and without PBAT film coatings is shown in the TGA results. A surface coating of PBAT induced an increase in weight loss to 7–13%. Noticeable weight loss in the paper samples was initially observed at 98.4 °C due to the evaporation of water that was anticipated to be consumed in the paper materials. The specimens showed their greatest weight losses at 278.6 and 372.0 °C. At 255.7 °C, the paper sample started to thermally decompose, and at 282.9 °C, the material coated with PBAT film started to decompose [55]. The paper’s maximum thermal degradation occurred at 365.4 °C, whereas the material coated with PBAT film attained 417.3 °C.

The DSC thermograms of uncoated and coated paper samples are presented in Figure 4B. When heating the original paper, weak endothermic effects were observed, probably due to the water evaporation and thermal decomposition of hemicelluloses. The endothermic peak at a temperature of 170.3 °C was related to the melting point of the PBAT film. Higher temperature values were reached with the PBAT film-coated sample endotherm. At lower temperatures, the transition began, and it concluded at higher temperatures. For the PBAT film-coated sample, both values were greatly raised. It is evident that the temperature was 133.8 °C, which is 90% higher than the first PBAT’s 115.3 °C and 45% lower, respectively.

### 3.4. Thickness and Tensile Strength

The thickness differed from 0.160 and 0.439 mm. As the PBAT coating was bi-layered with paper material, the film thickness was enhanced. This could be explained as an increase in crosslinking, causing the structure to be smaller and denser with a lower free volume, leading to a decrease in thickness. The results presented are similar to those of earlier research, which revealed that increased concentration of its function caused a decrease in the overall thickness of the PBAT film-coated paper.

Figure 5A displays the stress–strain curves for paper materials that were untreated and coated with PBAT film. The PBAT film-coated paper’s tensile strength (TS) and elongation at break (EAB) were calculated and are shown in Figure 5B. Tensile strength (TS) and elongation at break (EAB) are the two mechanical characteristics that should be studied in order to identify the packing materials’ strength. It was determined that the PBAT film coating had a major effect on the paper’s TS and EAB. The paper’s grammage increased when the coating was uniform. The bonding of the PBAT film was attributed to an increase in paper grammage. The TS and EAB of the paper without a coating were found to be 28.4 ± 4.77 MPa and 152.13%. The presence of the -C=O group on the PBAT was connected with the high TS values of the coated paper, which assisted in the mobility of the molecules and resulted in increased TS. When the PBAT film was coated onto paper, the samples had a maximum TS of 53.3 ± 5.10 MPa and a minimum EAB of 521.82%.

The smooth coating of PBAT was more effective in terms of increasing TS and EAB. The attraction between the PBAT was the main reason for the enhancement in tensile properties. This can also be attributed to an increased contact area between the PBAT and paper material. The uniform and even PBAT film coating on the paper can be clarified by studying its bursting strength. The fabrication process, the beating and refining process, the PBAT’s quality, and the addition of materials all tend to have an effect. Illustrated in Figure 5C are the burst strength and burst index of the paper both coated and uncoated with PBAT film. For the paper without PBAT coating, the values were 0.71 kPa m^2^/g and 58.2 kPa for burst strength and burst index, respectively. After the coating was uniform, these values were gradually enhanced. Following an even coating of PBAT, the values of the burst strength and burst index were 1.55 kPa m^2^/g and 119.1 kPa. In terms of burst strength, there must be good interface bonding with the PBAT film and the paper. Furthermore, when the coating of PBATs is uniform, it also increases the TS’s overall developmental actions, whereas EAB increases.

### 3.5. Porosity and Water Absorption Value

The porosity and water absorption values of the PBAT-film-coated and -uncoated papers are shown in Figure 6A. The porosity of the uncoated paper materials was 85.4 mL/min. After coating the PBAT film onto the paper, the amount decreased significantly to 16.7 mL/min. Shankar and Rhim [46] reported that the coatings caused the film voids in the paper to enlarge, decreasing its porosity. The Cobb and Cobb index values for uncoated paper were 51.19 g/m^2^ and 0.653, respectively (Figure 6B). For PBAT film-coated paper, the Cobb and Cobb index values were 26.20 g/m^2^ and 0.334, respectively.

The water absorbency of the PBAT film-coated papers decreased slightly, but not greatly. A value was calculated for the PBAT film coatings in the paper. It was reasonable, given the hydrophilic characteristics of the paper. Also, it was demonstrated that the PBAT film coatings reduced their capacity to absorb water. The coatings of PBAT film may be the reason, as they provided a non-polar surface. In addition, the backbone of PBAT contained CH_2_, C-O, and C=O groups. As a result, PBAT was hydrophobic and had a high nonpolar property, and could react with water to generate another hydrophobic material.

### 3.6. Barrier Properties

The permeation of the materials used for packaging has a main effect on the shelf-life of packaged goods. Figure 6C shows the OTR and WVTR of the PBAT film-coated paper material. The paper without coating had a transmission rate of 2010.40 cc m^−2^ per 24 h for OTR and 110.24 g m^−2^ per 24 h for WVTR. The value dropped to 992.86 cc m^−2^ per 24 h and 91.79 g m^−2^ per 24 h if the PBAT film was coated. The OTR and WVTR measurements were dropped using PBAT film [56]. This resulted in a taut surface structure of the paper, which was produced by the pressure used during drying. In addition, the permeability of water vapor and oxygen was decreased due to a reduction in porosity. The permeability of OTR and WVTR was reduced by a bi-layer of PBAT on the paper.

### 3.7. Water Contact Angle Analysis

Water angle contact is generally used to evaluate surface wettability. For uncoated paper, the contact angle (after 1 s) presented a lower value (67.40°), which was expected due to the hydrophilic nature of the cellulose and the porous structure of the paper [57,58]. After 120 s, the drop spread over the hydrophilic surface of the paper, decreasing its value to 30.19°. Nevertheless, for the coated paper, the value of the contact angle increased to 96.26° (that did not change with time), which clearly indicated presence in the surface of the hydrophobic PBAT, as can be seen in Figure 7. This coating process was more efficient compared to the work of Shankar et al. who found no improvement in the water contact angle for the PBAT-coated paper [49]. The water absorption capacity of the paper surface was also determined using a method based on the Cobb test for 120 s. The amount of water absorbed by the uncoated paper during 120 s was 250 g/m^2,^ while the PBAT-coated paper absorbed no water. Together with the contact angle results, this suggests that the porosity of the paper is greatly reduced when coated with PBAT, resulting in paper with high resistance to wettability and water penetration.

### 3.8. Food Quality Test of Tomatoes

Food quality is currently a major concern for consumers who demand both high-grade products and attractive items. On the other hand, if the food is not consumed directly, certain components (bacteria, air, moisture, and light) can accelerate its decomposition. If fruits and vegetables attain an additional level of maturity—that is, once they start to lose their firmness, exhibit wrinkles, or change color—they exhibit this phenomenon. As a result of these two procedures, food is wasted, as it is considered to be unsuitable for consumption by customers, especially in recent years [59]. After the packing of the tomatoes in PBAT film-coated paper packaging material, uncoated paper, and control, the characteristics of the tomatoes were examined. Figure 8 shows images of the tomatoes’ physical appearance when packaged with paper coated with PBAT film, paper without coating, and a control sample. When tomatoes were packaged in paper material, their weight was reduced instantly, while the weight of the tomatoes covered with the PBAT film-coated paper samples dropped more slowly.

Table 2 displays the weights of the tomatoes that were placed within the paper coated with PBAT film and the control sample over two weeks. Three classifications were utilized to classify the sensory character outcomes: satisfactory, average, and unacceptable. If the tomatoes were placed under the paper coated with PBAT film, their physical properties changed. Tomatoes became softer, their color changed from bright red to pale red, their smell became less pleasant, and their firmness reduced. Table 3 evaluates tomatoes wrapped in paper packaging coated with PBAT film from day 1 to day 14. Physical characteristics like weight, color, firmness, and smell are taken into consideration.

## 4. Conclusions

The low-cost, highly effective biobased PBAT film holds immense potential for various applications in packaging. In this study, we explored the application of PBAT film coatings for paper material. Comprehensive analyses, including FTIR, XRD, SEM, TGA, and DSC, were conducted to assess the coating efficiency achieved through hot pressing. Coating the paper with PBAT film resulted in a significant increase in the tensile strength (TS) and elongation at break (EAB), rising from 28.4 MPa to 53.3 MPa and from 152.13% to 521.82%, respectively. Notably, the oxygen transmission rate (OTR) was effectively controlled by the PBAT film coating, ranging from 2010.40 cc m^−2^ per 24 h to 992.86 cc m^−2^ per 24 h. Moreover, the water vapor transmission rate (WVTR) of the PBAT-coated paper decreased to 91.79 g m^−2^ per 24 h. These results highlight the potential of PBAT as an environmentally friendly alternative to conventional petroleum-based materials. Real-time shelf-life testing demonstrated that the tomatoes remained acceptable until day 14 when coated with PBAT film, in contrast to control packing materials, which showed deterioration by day 6 and disintegration by day 9. The primary goal of this research is to mitigate postharvest losses of tomatoes, particularly in settings with inadequate cold storage or processing facilities. In conclusion, this study affirms the overall excellent performance of PBAT film-coated paper material. The use of PBAT film-coated paper significantly extends the shelf-life of tomatoes from 1 to 14 days compared to control and uncoated paper. Therefore, we recommend the application of PBAT film-coated paper in packaging to prolong the shelf life and maintain the quality of tomatoes.

## Figures and Tables

**Figure 1 polymers-16-01283-f001:**
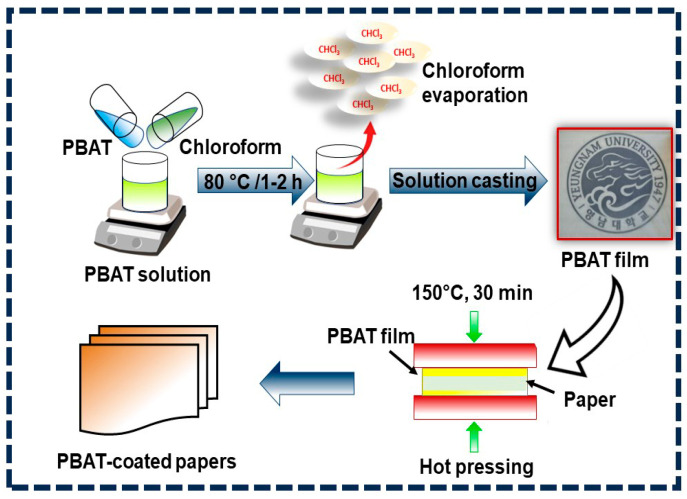
The process for placing PBAT coating onto the paper surface.

**Figure 2 polymers-16-01283-f002:**
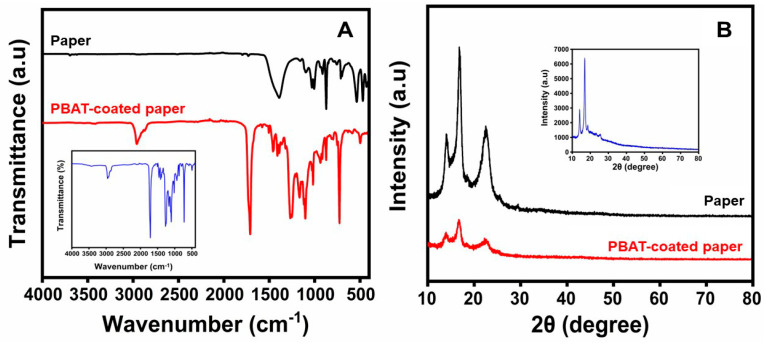
(**A**) FTIR spectra; (**B**) XRD pattern of the paper and PBAT-coated paper. PBAT film is shown in (**A**,**B**) [inset] for comparison. Additionally, at nearly 2964 cm^−1^, two absorption bands were observed concerning the symmetric and asymmetric vibrations of aliphatic C–H bonds. The PBAT and the in-plane bending vibration of CH_2_ bonds within the 1455 cm^−1^ and 1412 cm^−1^ region were related to the C=O-stretching bands of polyester, which were noticed at 1710 cm^−1^. In accordance with a study by Pietrosanto et al. [48], the benzene ring’s out-of-plane deformation on the PBAT could be changed by the bands at 720 cm^−1^. The crystalline or amorphous structures of the PBAT film-coated and uncoated paper material studied with X-ray diffraction (XRD) are shown in Figure 2B. Four diffraction angles were observed in the PBAT film-coated papers [49]; these corresponded to the 2θ at 13.95°, 16.67°, 23.19°, and 25.29°. The peaks at 16.67° and 23.2° show the PBAT structure’s distinct diffract angles. According to the paper, the diffraction angle should be seen at 2θ of 22.52°, where the PBAT peak overlaps. The characteristic crystalline peak of PBAT at 2θ = 13.98° seems too weak to be evident in the material used for packaging in the current study.

**Figure 3 polymers-16-01283-f003:**
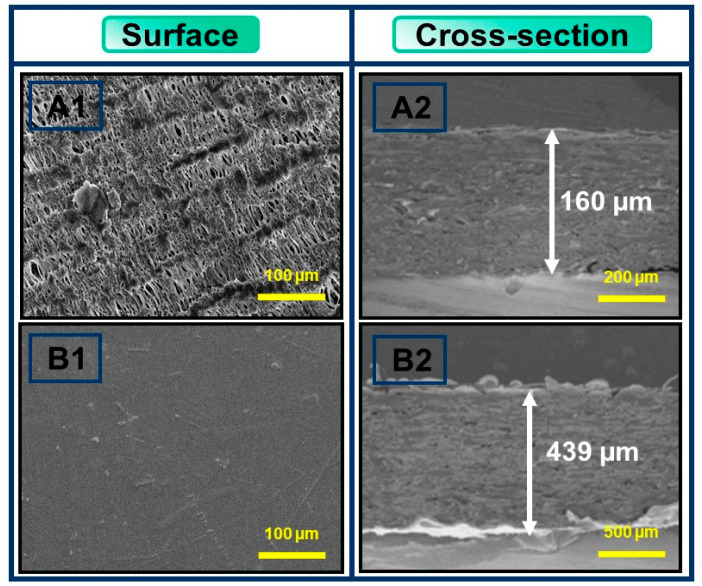
SEM images of the surface and cross-section of (**A1**,**A2**) paper and (**B1**,**B2**) PBAT-coated paper material.

**Figure 4 polymers-16-01283-f004:**
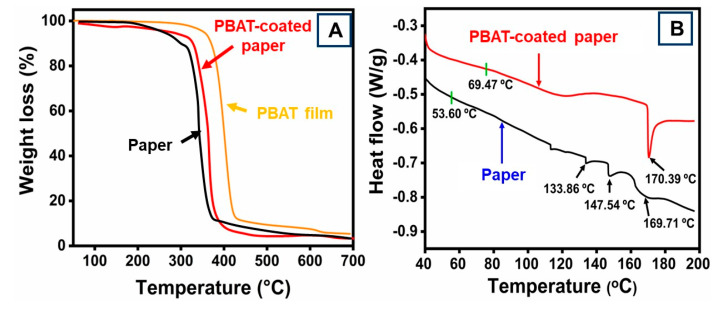
(**A**) TGA curve and (**B**) DSC curves of paper and PBAT-coated paper. The TGA curve of PBAT film insert (**A**).

**Figure 5 polymers-16-01283-f005:**
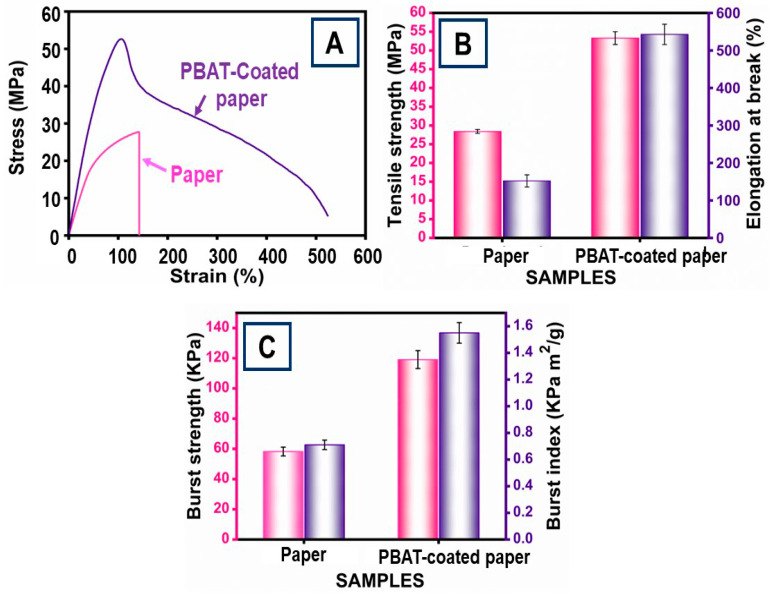
(**A**) Stress–strain curves; (**B**) tensile strength and elongation at break; (**C**) burst strength and burst index values of paper and PBAT-coated paper: The error bars show ± 5.00 standard errors.

**Figure 6 polymers-16-01283-f006:**
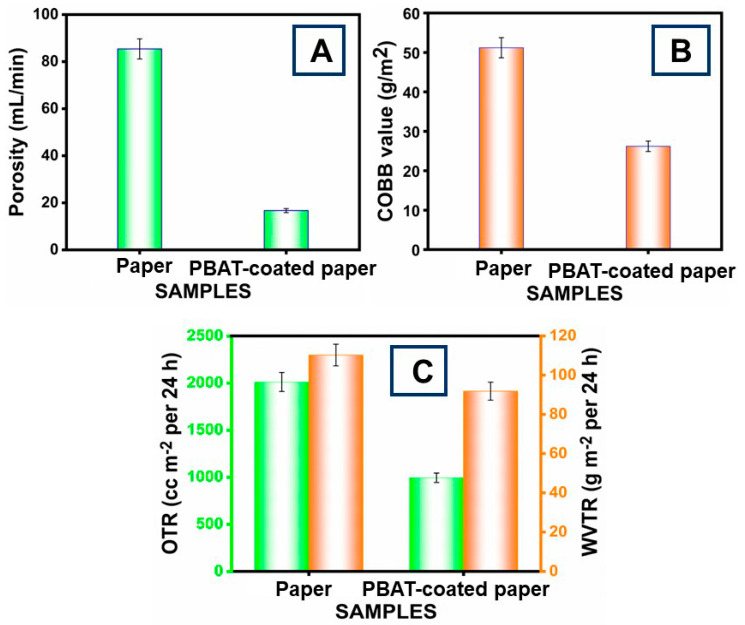
(**A**) Paper porosity values decreased with PBAT coating; (**B**) water absorption values reduce as PBAT film coated the paper; (**C**) barrier properties of PBAT-coated and uncoated paper material. The error bars show ±5.00 standard errors.

**Figure 7 polymers-16-01283-f007:**
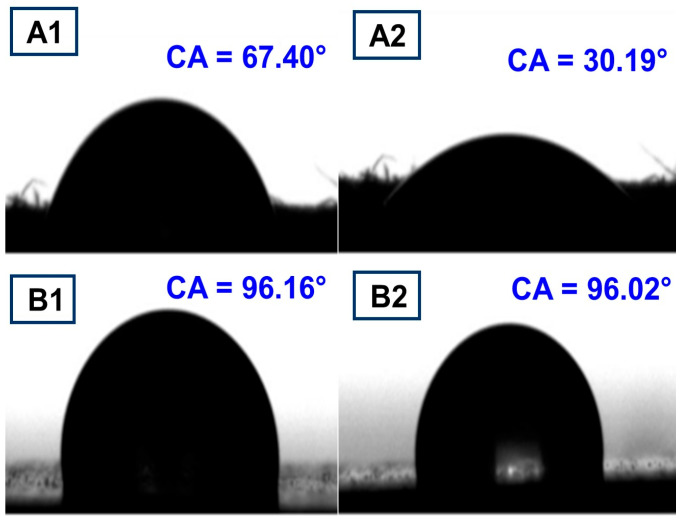
Images of water contact angle after 1 s and 120 s of the papers (**A1**,**A2**) and PBAT-coated paper (**B1**,**B2**).

**Figure 8 polymers-16-01283-f008:**
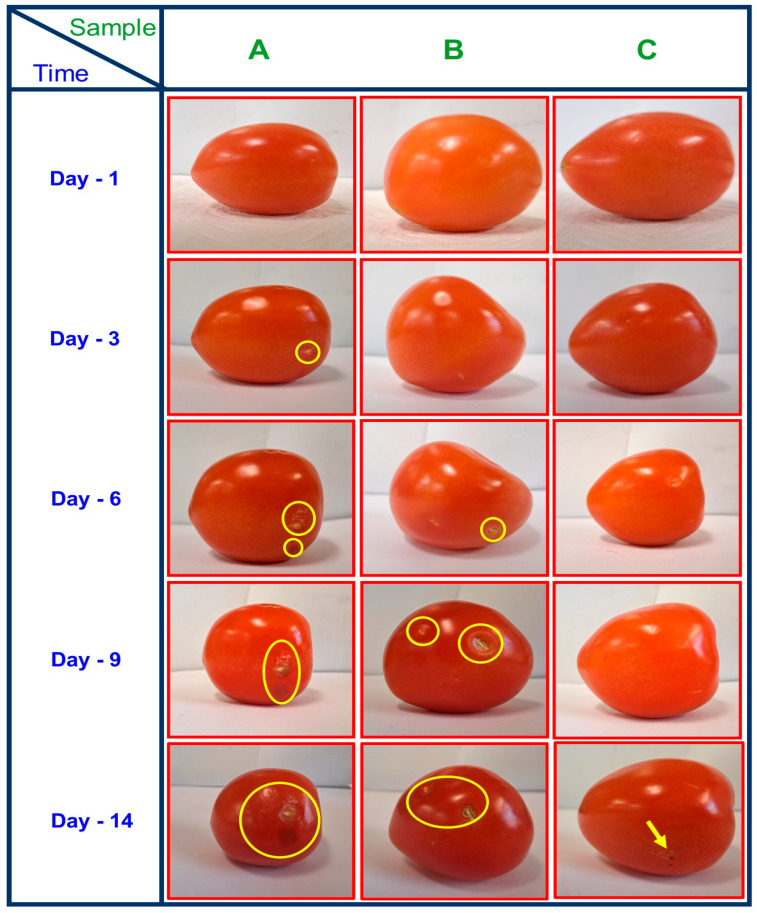
The appearance of the tomatoes covered with (**A**) control (open air), (**B**) uncoated paper, and (**C**) paper coated with PBAT film during storage at room temperature.

**Table 1 polymers-16-01283-t001:** Properties of the paper, coated and uncoated.

S. No.	Properties	Paper	PBAT-Coated Paper
1.	Grammage (g/m^2^)	180 ± 5.1	197.0 ± 9.0
2.	Thickness (µm)	160 ± 2.0	439.4 ± 6.0
3.	Density (g/cm^3^)	0.645	0.916
4.	Bendtsen permeability (mL/min)	130 ± 20	0
5.	Water angle contact (°)	72.3 ± 1.0	96.1 ± 1.2

**Table 2 polymers-16-01283-t002:** Weight loss of the tomatoes packed with paper, and paper coated with PBAT.

S. No.	Days	Weight Loss (g)		
		Control (Open Air)	Paper	PBAT Coated Paper
1.	1	79.437	72.621	80.534
2.	3	76.454	70.079	80.102
3.	6	75.722	69.612	79.708
4.	9	73.098	67.887	79.091
5.	14	62.192	57.220	75.891

**Table 3 polymers-16-01283-t003:** Comparing the physical parameters of the tomatoes on day 1 and day 14.

S. No.	Parameters	Day 1	Control (Open Air) on Day 14	Paper on Day 14	Coated Paper on Day 14
1.	Weight (g)	Control (open air)—79.437	62.192	57.220	75.891
		Paper—72.621			
		Coated paper—80.534			
2.	Color	Dark red	Pale red	Pale red	Pale red
3.	Odor	Satisfactory	Malodorous	Average	Acceptable
4.	Firmness	Firm	Soggy	Soft and soggy	Soft

## Data Availability

Data are contained within the article.

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
