# Peer review of "Eco-Friendly Poly (Butylene Adipate-co-Terephthalate) Coated Bi-Layered Films: An Approach to Enhance Mechanical and Barrier Properties"

_polymers, 2024, doi:10.3390/polym16091283_

Round 1

Reviewer 1 Report

Comments and Suggestions for Authors

The submitted manuscript proposes a new packaging for food and other goods, which is created by double-sided application of PBAT film on paper board. This relatively simple packaging material is extensively and comprehensively characterized, including a study for the preservation of a real plant product. The authors describe the current issues in a readable manner, detailing the procedures and methods used, the analytical techniques, and logical descriptions of the measurement results. The result is a new packaging design that has confirmed suitable properties for further uses and applications.

The only major comment is the thickness of the PBAT film. Firstly, the authors talk about coating, but the film itself is 3 times thicker than paper board (see Table 1). Here it is not clear whether this is the overall thickness or the thickness of one side. Given these proportions, one cannot speak of a coating, but rather a sandwich, composite, or similarly named structure. The first sentence on line 266 indicates a change in thickness, yet it is not clear which thickness is meant. That is, the description of the thickness parameter is messy and must be clearly defined in all steps. Secondly, the thickness of the film (on one or both sides of the carton) is a parameter that determines the properties of the packaging, the application, etc. and is therefore quite critical. How did the authors arrive at the thickness used? It would be useful to optimize this parameter in terms of a selected application property, such as oxygen permeability or other. Please, at the very least, comment on this with appropriate reasoning. After completing the manuscript according to the above recommendations, I can agree with the publication.

Comments on the Quality of English Language

The abbreviation PBAT is not appropriate for the title of the publication, the name "coating" is misleading, manufacturers including state and city are not listed for some instruments, and typographical errors of units especially in the abstract.

Author Response

Comment 1.: The submitted manuscript proposes a new packaging for food and other goods, which is created by double-sided application of PBAT film on paper board. This relatively simple packaging material is extensively and comprehensively characterized, including a study for the preservation of a real plant product. The authors describe the current issues in a readable manner, detailing the procedures and methods used, the analytical techniques, and logical descriptions of the measurement results. The result is a new packaging design that has confirmed suitable properties for further uses and applications.

Response: We are thankful to the reviewer for insightful comments and suggestions and express gratitude for the time spent reviewing our manuscript. We have modified the manuscript in lieu of the reviewer’s comments.

Comment 2.: The only major comment is the thickness of the PBAT film. Firstly, the authors talk about coating, but the film itself is 3 times thicker than paper board (see Table 1). Here it is not clear whether this is the overall thickness or the thickness of one side. Given these proportions, one cannot speak of a coating, but rather a sandwich, composite, or similarly named structure. The first sentence on line 266 indicates a change in thickness, yet it is not clear which thickness is meant. That is, the description of the thickness parameter is messy and must be clearly defined in all steps. Secondly, the thickness of the film (on one or both sides of the carton) is a parameter that determines the properties of the packaging, the application, etc. and is therefore quite critical. How did the authors arrive at the thickness used? It would be useful to optimize this parameter in terms of a selected application property, such as oxygen permeability or other. Please, at the very least, comment on this with appropriate reasoning. After completing the manuscript according to the above recommendations, I can agree with the publication.

Response: We are thankful to the reviewer for insightful comments and suggestions and express gratitude for the time spent reviewing our manuscript. We have modified the manuscript in lieu of the reviewer’s comments.

In fact, provided that we use a bi-layer coating on a paper surface. The obtained PBAT film has a thickness value between 120 and 150 μm.

The goal of this paper is to package materials onto coated paper via hot pressing.  for the reason we referred to as "coating".

The revised manuscript provides enhanced paper thicknesses and coated paper according to reviewer comments. We are using bi-layering coated paper with the help of the hot-pressing process. There is a uniform coating on both sides of the coated paper. The SEM test and thickness measurements confirm the surface coating.

The manuscript had been smoothed for grammatical, format, and errors, and the revised version of the manuscript was corrected by a native English speaker.

Comment 3.: The abbreviation PBAT is not appropriate for the title of the publication, the name "coating" is misleading, manufacturers including state and city are not listed for some instruments, and typographical errors of units especially in the abstract.

Response: The PBAT abbreviation is mentioned in the title, per reviewer comments. The name "coating" has been revised and changed, and the manufacturer's state and city details has been provided. Finally, a second review of the whole paper was carried out with the goal to correct any grammatical and typographical errors.

Eco-Friendly Poly (Butylene Adipate-Co-Terephthalate) Coated Bi-Layered Films: An Approach to Enhance Mechanical and Barrier Properties

Reviewer 2 Report

Comments and Suggestions for Authors

After studying this manuscript, many shortcomings were discovered.

General Remak: As can be understood from further results, the problem is that the authors used the wrong terminology when calling thin paper paperboard. In addition, in the “Materials,” the authors should indicate that the original substrate is thin paper with a grammage of 45.1 g/m2, and not a paperboard with a grammage of 250 g/m2. Further, in the manuscript text the term “paperboard” must be replaced with the term “paper”.

Title: Enhanced Mechanical and Barrier Properties of Paperboard with Eco-Friendly PBAT Film Bi-Layered Coating. Remark: Since the initial substrate was paper, the title should be corrected as:  Enhanced Mechanical and Barrier Properties of Paper with Eco-Friendly PBAT Film Bi-Layered Coating.

Abstract. Remark: Replace the term “paperboard” with the term “paper” in the Abstract.

Introduction

Remark: Replace the term “paperboard” with the term “paper” in the Introduction.

Lines 34-37. The limitation of paper and paperboard as packaging is a high level of porosity, which decreases barrier characteristics... Remark: An additional limitation is the hydrophilicity of the paper.

Lines 36-38. Regarding the replacement of petroleum-based packaging material with bio-based packaging material. Remark: There is no connection with the previous sentence. On the contrary, this statement contradicts the previous one, since just paper and paperboard is a bio-based packaging materials that does not need to be replaced by another bio-based material. Therefore, this sentence should be removed.

Line 55. Remark: It is not clear why PBAT and not another hydrophobic polymer was chosen for paper coating. Since PBTA has poor mechanical properties, high production costs, and limited applications (lines 63-64), what criteria did the authors use when choosing PBAT? Was it because PBAT is biodegradable? If yes, the reference or references about the biodegradability of this polymer should be added. In addition, the authors must provide evidence that this polymer has advantages over other biodegradable polymers, for example over EcoPla (Cargill Dow Polymers), aliphatic polylactides or polyamides, etc.

Line 55. PBAT is one of the hydrophobic biobased materials... Remark: Why do the authors consider that PBAT is biobased if the monomers - butylene adipate and terephthalate, used for the synthesis of this polymer are produced from fossil sources - gasoline or natural gas. Seems that the word “biobased” should be removed.

Line 56. Polymerization of ... produce PBAT. Remark: PBAT is produced by polycondensation reaction and not by polymerization.

Lines 57-58. Its (PBAT) remarkable strength and flexibility... Remark: This statement contradicts to “poor mechanical properties” of PBTA discovered in lines 63-64. Therefore, the sentence in lines 55-58 should be corrected or removed.

Lines 67-68. The tensile strength of PBAT materials has been increased with its application. Remark: This statement is meaningless; moreover, it contradicts to “PBAT’s poor mechanical properties ... limit its commercial application” (lines 63-64).

Lines 71-73. To avoid this, the paper (board) needs to be coated with polymers...  Remark: It is unclear what needs to be avoided? “to enhance its overall performance and reduce its cost of usage”? (lines 69-70). The sentence in lines 71-72 has no relation to the previous sentence and should be corrected.

Materials and Methods

2.2 Fabrication of PBAT film-coated paper. Remark: The method of producing PBAT film by casting a PBAT solution onto a glass plate is irrational, uneconomical, unproductive, harmful, and generally unnecessary. The authors also note that chloroform is a harmful solvent and PBAT is difficult to dissolve in it; therefore, the use of PBAT solution for film formation is not considered cost-effective (lines 181-183).

The simplest, most productive, and direct method used in the industry is to melt PBAT granules and then coat the surface of the cardboard with this melt to form a thin film on the surface of the substrate.

Line 115.  (SEM, Hitachi S-4800, ... Remark: Move the bracket, as follows, “SEM (Hitachi S-4800, ...

Lines 118-126. 2.3.3. Thermal Characterization. Remark: The description is unclear and contradictory, therefore it should be corrected, e.g., as follows, “To test thermal stability, a thermogravimetric analyzer SDT Q600 of TA Instruments) was used. In the case of TGA experiments, the samples were heated in an N2 atmosphere at a rate of 10 °C/min up to 700oC. In the case of DSC experiments, the samples were preheated to 180oC in an N2 atmosphere, kept for 2 min, and then cooled. Further, the samples were heated in N2 flow with a rate of 20°C/min up to 300°C”.

Results

Fig. 1. The process for placing PBAT-film coating to paperboard surface. Remark: Replace “paperboard” with “paper” surface.

Table 1.  Properties of the paperboard, coated and uncoated. Remark: Replace “paperboard” with “paper”.

Additional Remark: The presented in Table 1 specifications are incorrect. In the “Materials” section, the original duplex paperboard has a grammage of 250 g/m2, while Table 1 states that the original uncoated paperboard has a grammage of only 45.1 g/m2, which is typical for very thin paper, not for paperboard. In addition, it is known that the thickness of a typical paperboard is more than 0.3 mm, not 0.14 mm (140 μm) as shown in Table 1. Since in Table 1 data for thin paper is given instead of the original paperboard, the data for coated paperboard is also incorrect. It is obvious that the authors presented data for a paper substrate such as very thin paper. If the authors made an error in the “Materials” and the original material is not paperboard but thin paper having a grammage of 45.1 g/m2, then this error must be corrected both in the “Materials” and in Table 1.

As can be understood from further results, the problem is that the authors used the wrong terminology when calling thin paper paperboard. Then in the “Materials,” the authors should indicate that the original substrate is thin paper with a grammage of 45.1 g/m2, and not paperboard with a grammage of 250 g/m2. Further, in the all manuscript text the term “paperboard” must be replaced with the term “paper”.

Fig. 2. (A) FTIR spectra; (B) XRD pattern of the paperboard, and PBAT film-coated paperboard. PBAT film is shown in (A) and (B) [inset] for comparison. Remark: Replace “paperboard with “paper”

Fig. 2(A) represents the functional group of the paper (board) and PBAT film-coated paper (board). Remark: This sentence is incorrect since Fig. 2(A) represents the FTIR spectra and not the functional group. Thus, correct this sentence as follows, Fig. 2(A) represents the FTIR spectra of uncoated and coated paper samples”.

Figures 3-9. Remark: In the names of these figures replace “paperboard with “paper”

Fig. 4. SEM micrographs of surface and cross-section of (A1, and A2) paper, (B1, 234 and B2) PBAT-coated paper material. Remark: This Figure shows that the thickness of the uncoated substrate is 140 μm, while the thickness of the coated substrate is 305 μm.. Additional remark: Why the thickness of the coated substrate in Figure 4 is 305 μm, and in Table 1 is 419.4 μm? What thickness value is correct?

Line 266. The thickness differed from 0.411 and 0.147 mm. Remark: This data is incorrect. 

The English language also needs to be corrected.

Lines 52-53. Polymers have been shown to be ... Remark: “have been to be” should be replaced with “are”.

Line 56. Polymerization ..... produce... Remark: Use here a singular form of the verb “produces” instead of the plural form “produce”.

Lines 59-60. According to Rajendran and Han, [19]; and Naser et al., 59 [20], it is therefore believed to be the packaging material which may replace polyethylene... Remark: This sentence must be corrected e.g., as follows, “Rajendran and Han, [19]; and Naser et al., [20], believe that this polymer material can efficiently replace polyethylene”

Lines 72-73. ...also extend the shelf-life of food's [28, 29]. Remark: Correct this phrase, as follows, ...”also extending the shelf-life of food”

Lines 75-76. ...have a less quality than virgin fibers (due to the drying process of the recycled paperboard leads the fiber size... Remark: Replace “less” with “lower” and add “to” before the fiber size...

Line 78. ...via hot pressing... Remark: Replace “via” with “by”.

Line 80. to a 150 °C Remark: Remove the article “a”.

Line 81. ... render it... ... Remark: Write “rendering” instead of “render”

Lines 93-94. ...prior usage... Remark: Add “to” before “usage”.

Lime 96. Prior to use, Remark: Replace “Prior to” with “Before”

Line 120. In a N2 atmosphere... Remark: Replace article “a” with “an”

Line 122. ...used 10.0 mg of material from all of each of the paperboard... Remark: Correct this phrase as follows, ...”10.0 mg of each sample were used” ...

Lines 128-129. The aim of working with the estimated values was to identify the mechanical property of coated and uncoated paperboard. Remark: This sentence should be corrected as follows, “In this research, the mechanical properties of coated and uncoated paperboard were tested”.

Line 129. ...characteristics evaluated... Remark: The verb “were” should be added, namely, ...” characteristics were evaluated” ...

Line 130. the TAPPT T494... [38].  Remark:  Write “TAPPI” instead of “TAPPT”.

Lines 130-132. To evaluate the sizes of both coated and uncoated paperboard specimens, 25 × 180 mm of paperboard material had been cut with the paperboard's cellulose fibers. Remark: This sentence should be corrected as follows, “The specimens for the TAPPI T494 test [38]with sizes of 25 × 180 mm were prepared by cutting both coated and uncoated paperboard”.

Line 136. ...it is also known... Remark: Remove “is”.

Lines 136-137. Specimen cutter was used to cut the paper specimens into 100 × 100 mm sizes in accordance with TAPPI T403 [39] standard. Remark: This sentence should be corrected as follows, “To cut specimens into 100 × 100 mm sizes for TAPPI T403 test [39], a special curter was used”.

Lines 138-139. Bursting strength, which was measured and mentioned in KPa, is the highest pressure reading up to the rupture point. Remark: This sentence should be corrected as follows, “Bursting strength measured in kPa represents the highest strength value achieved before rupture”.

 2.3.5. Porosity and Water Absorption (COBB). Remark: Remove (COBB) from this title

Additional remark: The text of 2.3.5 contains grammatical errors and should be corrected, e.g., as follows, “The Cobb-60 method was used to measure the water absorption capacity of uncoated and coated paperboard expressed in gH2O per m2 of sample. These measurements were carried out using a COBB tester (Test Techno, India) at room temperature by the TAPPI T456 standard procedure [40]. Samples with the size of 10 x 10 cm were conditioned at 23 ± 1 °C and 50 ± 1% relative humidity. Dividing the Cobb value by grammage yielded the Cobb index.

The porosity of the samples was measured using a Frank PTI porosimeter by the TAPPI T 460 standard method. For this purpose, samples with a diameter of 100 mm were prepared. The porosity test of dry samples was carried out at 23 °C under pressure of 1.47 kPa”.  

Line 156. method was used... Remark: Replace “was with “were”.

Lines 157-158. The measurements were achieved several times before averaging the results and tests were performed at RT. Remark: This sentence should be corrected as follows, “The measurements were performed several times to calculate an average result”. Remove the final phrase “and tests were performed at RT”.

Lines 160-164. Remark: The text of 2.3.7 contains grammatical errors and should be corrected, e.g., as follows, “OCA-20 of Dataphysics Instruments was used to determine the contact angle of uncoated and coated paperboard samples. Water drops of 1 μl volume were used for these measurements. The image of the drops was captured for 5 seconds.

Lines 165-168. Remark: The text of 2.3.8 contains grammatical errors and should be corrected, e.g., as follows, “The received results were analyzed by the variation ANOVA method using the SPSS statistical software package (SPSS Thailand Ltd., Thailand). The value of P was less than 0.05. To identify significant changes Tukey's multiple evaluation test was used”.

The received results were analyzed by the variation ANOVA method using the SPSS statistical software package (SPSS Thailand Ltd., Thailand). A P value was less than 0.05. To identify significant changes Tukey's multiple evaluation test was used.

Lines 172-173.  Applying two layers of PBAT film and confirmed the paperboard surface was without spot, the uniform coatings had been confirmed [43]. Remark: Correct this sentence, as follows, “After applying PBAT films on paperboard surfaces, the coating uniformity was confirmed [43].

Lines 175-178. Since the uniform coating was confirmed, the outcomes recommended that the coating methods may entered into the fibers, which may have served to improve the barrier properties, etc. Remark: Introduce corrections, as follows, “The results showed that the polymer coating may enter the fibers, which can improve the barrier properties of coated paperboard. It is conceivable that the coating with BPAT films contributed to the superior uniformity results compared to those described by Hashmi et al., [45].

Lines 181-183. However, as PBAT is hazardous to chloroform and is difficult to dissolution in less hazardous solvents, with PBAT as a solution is not considered to be a cost-effective. Remark: This sentence contains grammatical errors and should be corrected, as follows, “However, the chloroform is a hazardous solvent and PBAT is difficult to dissolve in it; therefore, using PBAT solution for film forming is not considered cost-effective”.

Line 192. ...occur nearby 3424 cm−1, ... Remark: Remove “by” and write “... occur near 3424 cm−1, ...”

Lines 201-202. ...with respect to the symmetric... Remark: Write “ concerning the symmetric”... instead of ...”with respect to...”

Lines 212-214. Whereas, the characteristic crystalline peak (2θ = 13.98°) for PBAT seems too weak to be evident in the material used for packaging in the current study due to plastic processing. Remark: Correct this sentence, as follows, “The characteristic crystalline peak of PBAT at 2θ = 13.98° seems too weak to be evident in the material used for packaging in the current study”.

Lines 217-218. Images from SEM can be used to exhibit when the PBAT coating affect the paperboard. Remark: Correct this sentence, as follows, “SEM images can be used to demonstrate the effects of PBAT coating on paperboard morphology”.

Line 223...similar results on the smooth surface were obtained [50, 51]. Remark: Replace “on” by “with” as follows, “similar results with the smooth surface were obtained”.

Lines 224-225.  Whereas, there wasn't no vacant space in the paperboard's coating surface as the cellulose fiber pores  were fully filled with PBAT. Remark: Correct this sentence as follows, “However, there was no vacant space on the surface of the coated cardboard because the pores of the cellulose fibers were filled with PBAT”.

Line 245. ... induced the weigh loss to increase to 7% to 13%. Remark: Write instead, ...induced the increase in weight loss to 7- 13%.

Lines 245-246. The initial thermal degradation of the paperboard material was observed at 98.4°C due to water that... Remark: Correct this fragment, write “Noticeable weight loss in the paperboard samples was initially observed at 98.4°C due to evaporation of water that...”

Lines 252-258. Remark: This fragment contains many grammatical errors and needs major correction, as follows, “The DSC thermograms of uncoated and coated paperboard samples are presented in Fig. 5(B). When heating the original cardboard, weak endothermic effects were observed, probably due to the water evaporation and thermal decomposition of hemicelluloses. The endothermic peak at a temperature of 170.4oC is related to the melting point of the PBAT film”.  

ETC.

All text should be carefully checked and corrected.

Comments on the Quality of English Language

The English language is poor and needs to be corrected.

Lines 52-53. Polymers have been shown to be ... Remark: “have been to be” should be replaced with “are”.

Line 56. Polymerization ..... produce... Remark: Use here a singular form of the verb “produces” instead of the plural form “produce”.

Lines 59-60. According to Rajendran and Han, [19]; and Naser et al., 59 [20], it is therefore believed to be the packaging material which may replace polyethylene... Remark: This sentence must be corrected e.g., as follows, “Rajendran and Han, [19]; and Naser et al., [20], believe that this polymer material can efficiently replace polyethylene”

Lines 72-73. ...also extend the shelf-life of food's [28, 29]. Remark: Correct this phrase, as follows, ...”also extending the shelf-life of food”

Lines 75-76. ...have a less quality than virgin fibers (due to the drying process of the recycled paperboard leads the fiber size... Remark: Replace “less” with “lower” and add “to” before the fiber size...

Line 78. ...via hot pressing... Remark: Replace “via” with “by”.

Line 80. to a 150 °C Remark: Remove the article “a”.

Line 81. ... render it... ... Remark: Write “rendering” instead of “render”

Lines 93-94. ...prior usage... Remark: Add “to” before “usage”.

Lime 96. Prior to use, Remark: Replace “Prior to” with “Before”

Line 120. In a N2 atmosphere... Remark: Replace article “a” with “an”

Line 122. ...used 10.0 mg of material from all of each of the paperboard... Remark: Correct this phrase as follows, ...”10.0 mg of each sample were used” ...

Lines 128-129. The aim of working with the estimated values was to identify the mechanical property of coated and uncoated paperboard. Remark: This sentence should be corrected as follows, “In this research, the mechanical properties of coated and uncoated paperboard were tested”.

Line 129. ...characteristics evaluated... Remark: The verb “were” should be added, namely, ...” characteristics were evaluated” ...

Line 130. the TAPPT T494... [38].  Remark:  Write “TAPPI” instead of “TAPPT”.

Lines 130-132. To evaluate the sizes of both coated and uncoated paperboard specimens, 25 × 180 mm of paperboard material had been cut with the paperboard's cellulose fibers. Remark: This sentence should be corrected as follows, “The specimens for the TAPPI T494 test [38]with sizes of 25 × 180 mm were prepared by cutting both coated and uncoated paperboard”.

Line 136. ...it is also known... Remark: Remove “is”.

Lines 136-137. Specimen cutter was used to cut the paper specimens into 100 × 100 mm sizes in accordance with TAPPI T403 [39] standard. Remark: This sentence should be corrected as follows, “To cut specimens into 100 × 100 mm sizes for TAPPI T403 test [39], a special curter was used”.

Lines 138-139. Bursting strength, which was measured and mentioned in KPa, is the highest pressure reading up to the rupture point. Remark: This sentence should be corrected as follows, “Bursting strength measured in kPa represents the highest strength value achieved before rupture”.

 2.3.5. Porosity and Water Absorption (COBB). Remark: Remove (COBB) from this title

Additional remark: The text of 2.3.5 contains grammatical errors and should be corrected, e.g., as follows, “The Cobb-60 method was used to measure the water absorption capacity of uncoated and coated paperboard expressed in gH2O per m2 of sample. These measurements were carried out using a COBB tester (Test Techno, India) at room temperature by the TAPPI T456 standard procedure [40]. Samples with the size of 10 x 10 cm were conditioned at 23 ± 1 °C and 50 ± 1% relative humidity. Dividing the Cobb value by grammage yielded the Cobb index.

The porosity of the samples was measured using a Frank PTI porosimeter by the TAPPI T 460 standard method. For this purpose, samples with a diameter of 100 mm were prepared. The porosity test of dry samples was carried out at 23 °C under pressure of 1.47 kPa”.  

Line 156. method was used... Remark: Replace “was with “were”.

Lines 157-158. The measurements were achieved several times before averaging the results and tests were performed at RT. Remark: This sentence should be corrected as follows, “The measurements were performed several times to calculate an average result”. Remove the final phrase “and tests were performed at RT”.

Lines 160-164. Remark: The text of 2.3.7 contains grammatical errors and should be corrected, e.g., as follows, “OCA-20 of Dataphysics Instruments was used to determine the contact angle of uncoated and coated paperboard samples. Water drops of 1 μl volume were used for these measurements. The image of the drops was captured for 5 seconds.

Lines 165-168. Remark: The text of 2.3.8 contains grammatical errors and should be corrected, e.g., as follows, “The received results were analyzed by the variation ANOVA method using the SPSS statistical software package (SPSS Thailand Ltd., Thailand). The value of P was less than 0.05. To identify significant changes Tukey's multiple evaluation test was used”.

The received results were analyzed by the variation ANOVA method using the SPSS statistical software package (SPSS Thailand Ltd., Thailand). A P value was less than 0.05. To identify significant changes Tukey's multiple evaluation test was used.

Lines 172-173.  Applying two layers of PBAT film and confirmed the paperboard surface was without spot, the uniform coatings had been confirmed [43]. Remark: Correct this sentence, as follows, “After applying PBAT films on paperboard surfaces, the coating uniformity was confirmed [43].

Lines 175-178. Since the uniform coating was confirmed, the outcomes recommended that the coating methods may entered into the fibers, which may have served to improve the barrier properties, etc. Remark: Introduce corrections, as follows, “The results showed that the polymer coating may enter the fibers, which can improve the barrier properties of coated paperboard. It is conceivable that the coating with BPAT films contributed to the superior uniformity results compared to those described by Hashmi et al., [45].

Lines 181-183. However, as PBAT is hazardous to chloroform and is difficult to dissolution in less hazardous solvents, with PBAT as a solution is not considered to be a cost-effective. Remark: This sentence contains grammatical errors and should be corrected, as follows, “However, the chloroform is a hazardous solvent and PBAT is difficult to dissolve in it; therefore, using PBAT solution for film forming is not considered cost-effective”.

Line 192. ...occur nearby 3424 cm−1, ... Remark: Remove “by” and write “... occur near 3424 cm−1, ...”

Lines 201-202. ...with respect to the symmetric... Remark: Write “ concerning the symmetric”... instead of ...”with respect to...”

Lines 212-214. Whereas, the characteristic crystalline peak (2θ = 13.98°) for PBAT seems too weak to be evident in the material used for packaging in the current study due to plastic processing. Remark: Correct this sentence, as follows, “The characteristic crystalline peak of PBAT at 2θ = 13.98° seems too weak to be evident in the material used for packaging in the current study”.

Lines 217-218. Images from SEM can be used to exhibit when the PBAT coating affect the paperboard. Remark: Correct this sentence, as follows, “SEM images can be used to demonstrate the effects of PBAT coating on paperboard morphology”.

Line 223...similar results on the smooth surface were obtained [50, 51]. Remark: Replace “on” by “with” as follows, “similar results with the smooth surface were obtained”.

Lines 224-225.  Whereas, there wasn't no vacant space in the paperboard's coating surface as the cellulose fiber pores  were fully filled with PBAT. Remark: Correct this sentence as follows, “However, there was no vacant space on the surface of the coated cardboard because the pores of the cellulose fibers were filled with PBAT”.

Line 245. ... induced the weigh loss to increase to 7% to 13%. Remark: Write instead, ...induced the increase in weight loss to 7- 13%.

Lines 245-246. The initial thermal degradation of the paperboard material was observed at 98.4°C due to water that... Remark: Correct this fragment, write “Noticeable weight loss in the paperboard samples was initially observed at 98.4°C due to evaporation of water that...”

Lines 252-258. Remark: This fragment contains many grammatical errors and needs major correction, as follows, “The DSC thermograms of uncoated and coated paperboard samples are presented in Fig. 5(B). When heating the original cardboard, weak endothermic effects were observed, probably due to the water evaporation and thermal decomposition of hemicelluloses. The endothermic peak at a temperature of 170.4oC is related to the melting point of the PBAT film”.  

ETC.

All text should be carefully checked and corrected.

Author Response

After studying this manuscript, many shortcomings were discovered.

General Remak: As can be understood from further results, the problem is that the authors used the wrong terminology when calling thin paper paperboard. In addition, in the “Materials,” the authors should indicate that the original substrate is thin paper with a grammage of 45.1 g/m2, and not a paperboard with a grammage of 250 g/m2. Further, in the manuscript text the term “paperboard” must be replaced with the term “paper”.

Response: As per reviewer suggestions title has been changed. As per the reviewer's request, we have altered "paperboard" to "paper" in the whole corrected manuscript. On the recommendation of the reviewers, we double-checked the paper's GSM and thickness.

Paper grammage: 180 g/m2; thickness: 160 μm

Comment 1.: Title: Enhanced Mechanical and Barrier Properties of Paperboard with Eco-Friendly PBAT Film Bi-Layered Coating. Remark: Since the initial substrate was paper, the title should be corrected as:  Enhanced Mechanical and Barrier Properties of Paper with Eco-Friendly PBAT Film Bi-Layered Coating.

Response: As per reviewer suggestions title has been changed.

Eco-Friendly Poly (Butylene Adipate-Co-Terephthalate) Coated Bi-Layered Films: An Approach to Enhance Mechanical and Barrier Properties

Comment 2.: Abstract. Remark: Replace the term “paperboard” with the term “paper” in the Abstract.

Response: On the abstract, we have changed "paperboard" to "paper" according to the reviewer recommendation.

Comment 3.: Introduction; Remark: Replace the term “paperboard” with the term “paper” in the Introduction.

Response: "Paperboard" has been changed with "paper" per the reviewer's suggestion for the introduction section.

Comment 4.: Lines 34-37. The limitation of paper and paperboard as packaging is a high level of porosity, which decreases barrier characteristics... Remark: An additional limitation is the hydrophilicity of the paper.

Response: In the revised manuscript has added a comment indicating that the high level of hydrophilicity of the paper may decrease its barrier characteristics, as suggested by the reviewer.

However, paper materials can't be used as widely as they could be due to their high levels of hydrophilicity and porosity, which decrease their barrier characteristics.

Comment 5.: Lines 36-38. Regarding the replacement of petroleum-based packaging material with bio-based packaging material. Remark: There is no connection with the previous sentence. On the contrary, this statement contradicts the previous one, since just paper and paperboard is a bio-based packaging materials that does not need to be replaced by another bio-based material. Therefore, this sentence should be removed.

Response: The sentence has been removed and modified in accordance with reviewer comments.

The majority of studies have focused on the importance of replacement packaging fabricated from petroleum-based materials with alternatives to ensure the 21st century.

Comment 6.: Line 55. Remark: It is not clear why PBAT and not another hydrophobic polymer was chosen for paper coating. Since PBTA has poor mechanical properties, high production costs, and limited applications (lines 63-64), what criteria did the authors use when choosing PBAT? Was it because PBAT is biodegradable? If yes, the reference or references about the biodegradability of this polymer should be added. In addition, the authors must provide evidence that this polymer has advantages over other biodegradable polymers, for example over EcoPla (Cargill Dow Polymers), aliphatic polylactides or polyamides, etc.

Response: PBAT may have certain advantages in performance that make it suitable for application with paper coating. This may include its flexibility, adherence to paper, barrier characteristics, and compatibility with other coatings.

PBAT's has limited applications, high production costs, and low crystallization rate [22], could have considered additional factors onto consideration in choosing PBAT for paper coating. There are some things to reflect on: barrier characteristics [23], regulatory compliance, biodegradability [24, 25], suitability with paper recycling, market demand, and perception.

PBAT with other biodegradable polymers in relevant properties such as barrier properties, adhesion to paper, flexibility, printability, and durability. They would need to provide quantitative results or qualitative observations demonstrating on PBAT outperforms or meets the requirements better than the other polymers.

  1. Li, J.; Lai, L.; Wu, L.; Severtson, S.J.; Wang, W.-J. Enhancement of water vapor barrier properties of biodegradable poly(butylene adipate-co-terephthalate) films with highly oriented organomontmorillonite. ACS Sustain. Chem. Eng. 2018, 6, 6654–6662.
  2. Mujtaba, M.; Lipponen, J.; Ojanen, M.; Puttonen, S.; Vaittinen, H. Trends and challenges in the development of bio-based barrier coating materials for paper/cardboard food packaging; a review. Total Environ. 2022, 851, 158328.
  3. Adibi, A.; Trinh, B.M.; Mekonnen, T.H. Recent progress in sustainable barrier paper coating for food packaging applications. Org. Coat. 2023, 181, 107566.
  4. Lamsaf, H.; Singh, S.; Pereira, J.; Poças, F. Multifunctional properties of PBAT with hemp (Cannabis sativa) micronised fibres for food packaging: Cast films and coated paper. Coatings 2023, 13, 1195.

Comment 7.: Line 55. PBAT is one of the hydrophobic biobased materials... Remark: Why do the authors consider that PBAT is biobased if the monomers - butylene adipate and terephthalate, used for the synthesis of this polymer are produced from fossil sources - gasoline or natural gas. Seems that the word “biobased” should be removed.

Response: We removed the word "biobased" in due to reviewer comments.

PBAT may have certain advantages in performance that make it suitable for application with paper coating.

Comment 8.: Line 56. Polymerization of ... produce PBAT. Remark: PBAT is produced by polycondensation reaction and not by polymerization.

Response: We changed the "polymerization" to “polycondensation reaction” in due to reviewer comments.

PBAT is produced by polycondensation reaction using 1,4-butanediol, terephthalic acid, and adipic acid as raw materials, and using organic compounds as a catalyst.

Comment 9.: Lines 57-58. Its (PBAT) remarkable strength and flexibility... Remark: This statement contradicts to “poor mechanical properties” of PBTA discovered in lines 63-64. Therefore, the sentence in lines 55-58 should be corrected or removed.

Response: As suggested from the reviewer, we removed the "sentence" from lines 55–58.

Comment 10.: Lines 67-68. The tensile strength of PBAT materials has been increased with its application. Remark: This statement is meaningless; moreover, it contradicts to “PBAT’s poor mechanical properties ... limit its commercial application” (lines 63-64).

Response: Thank you for reviewer comments;

We carefully reviewed the PBAT polymers and noticed that it showed an excellent mechanical strength (25–35 MPa); thus, we dealt with the poor mechanical properties that were causing the low crystallization rate.

Comment 11.: Lines 71-73. To avoid this, the paper (board) needs to be coated with polymers...  Remark: It is unclear what needs to be avoided? “to enhance its overall performance and reduce its cost of usage”? (lines 69-70). The sentence in lines 71-72 has no relation to the previous sentence and should be corrected.

Response: We modified the sentence in accordance with the reviewer's suggestion.

This can be prevented by coated the paper with PBAT film, which can extend the shelf life of food and serve as a strong barrier against pollutants, water, and oils.

Comment 12.: Materials and Methods 2.2 Fabrication of PBAT film-coated paper. Remark: The method of producing PBAT film by casting a PBAT solution onto a glass plate is irrational, uneconomical, unproductive, harmful, and generally unnecessary. The authors also note that chloroform is a harmful solvent and PBAT is difficult to dissolve in it; therefore, the use of PBAT solution for film formation is not considered cost-effective (lines 181-183).

Response: We all agree with the reviewer's advice; thus, we removed the sentence (lines 181–183) in the revised manuscript.

Comment 13.: The simplest, most productive, and direct method used in the industry is to melt PBAT granules and then coat the surface of the cardboard with this melt to form a thin film on the surface of the substrate.

Response: Thank you for the comments; the hydraulic heating press method you described is a common industrial process for coating paper or cardboard with PBAT. This method, known as melt coating or extrusion coating, involves melting PBAT granules and then applying the molten polymer directly onto the surface of the cardboard substrate to form a thin film.

This method is preferred in industries due to its simplicity, productivity, and efficiency in coating large quantities of paper or cardboard substrates with PBAT. It allows for precise control over the thickness and uniformity of the PBAT coating, resulting in consistent quality and performance of the coated products. Additionally, it is a cost-effective process compared to other coating method.

Comment 14.: Line 115.  (SEM, Hitachi S-4800, ... Remark: Move the bracket, as follows, “SEM (Hitachi S-4800, ...

Response: As advised by the reviewers, we performed the needed changes.

Comment 15.: Lines 118-126. 2.3.3. Thermal Characterization. Remark: The description is unclear and contradictory, therefore it should be corrected, e.g., as follows, “To test thermal stability, a thermogravimetric analyzer SDT Q600 of TA Instruments) was used. In the case of TGA experiments, the samples were heated in an N2 atmosphere at a rate of 10 °C/min up to 700oC. In the case of DSC experiments, the samples were preheated to 180oC in an N2 atmosphere, kept for 2 min, and then cooled. Further, the samples were heated in N2 flow with a rate of 20°C/min up to 300°C”.

Response: We are agreed the reviewer comments; As advised by the reviewer, we performed the needed changes.

Comment 16.: Results; Fig. 1. The process for placing PBAT-film coating to paperboard surface. Remark: Replace “paperboard” with “paper” surface.

Response: "Paperboard" has been replaced to "paper" surface.

Comment 17.: Table 1.  Properties of the paperboard, coated and uncoated. Remark: Replace “paperboard” with “paper”.

Response: "Paperboard" was replaced with "paper".

Comment 18.: Additional Remark: The presented in Table 1 specifications are incorrect. In the “Materials” section, the original duplex paperboard has a grammage of 250 g/m2, while Table 1 states that the original uncoated paperboard has a grammage of only 45.1 g/m2, which is typical for very thin paper, not for paperboard. In addition, it is known that the thickness of a typical paperboard is more than 0.3 mm, not 0.14 mm (140 μm) as shown in Table 1. Since in Table 1 data for thin paper is given instead of the original paperboard, the data for coated paperboard is also incorrect. It is obvious that the authors presented data for a paper substrate such as very thin paper. If the authors made an error in the “Materials” and the original material is not paperboard but thin paper having a grammage of 45.1 g/m2, then this error must be corrected both in the “Materials” and in Table 1.

Response: We re-checked the paper's GSM and thickness according to recommendations from the reviewers. We have modified the properties of the paper in Table 1 and the Materials section.

Table 1. Properties of the paper, coated and uncoated.

S. No

Properties

Paper

PBAT-coated paper

1.

Grammage (g/m2)

180 ± 5.1

197.0 ± 9.0

2.

Thickness (µm)

160 ± 2.0

439.4 ± 6.0

3.

Density (g/cm3)

0.645

0.916

4.

Bendtsen permeability (mL/min)

130 ± 20

0

5.

Water angle contact (°)

72.3 ± 1.0

96.1 ± 1.2

Comment 19.: As can be understood from further results, the problem is that the authors used the wrong terminology when calling thin paper paperboard. Then in the “Materials,” the authors should indicate that the original substrate is thin paper with a grammage of 45.1 g/m2, and not paperboard with a grammage of 250 g/m2. Further, in the all-manuscript text the term “paperboard” must be replaced with the term “paper”.

Response: We changed "Paperboard" to "paper" based on reviewer comments.

Paper grammage: 180 g/m2

Paper thickness: 160 μm

Comment 20.: Fig. 2. (A) FTIR spectra; (B) XRD pattern of the paperboard, and PBAT film-coated paperboard. PBAT film is shown in (A) and (B) [inset] for comparison. Remark: Replace “paperboard with “paper”

Response: We changed "paperboard" to "paper" in Fig. 2(A) and (B) in response to reviewer comments.

Comment 21.: Fig. 2(A) represents the functional group of the paper (board) and PBAT film-coated paper (board). Remark: This sentence is incorrect since Fig. 2(A) represents the FTIR spectra and not the functional group. Thus, correct this sentence as follows, “Fig. 2(A) represents the FTIR spectra of uncoated and coated paper samples”.

Response: As advised by the reviewer, we performed the needed changes.

Fig. 2(A) represents the FTIR spectra of uncoated and coated paper samples.

Comment 22.: Figures 3-9. Remark: In the names of these figures replace “paperboard with “paper”.

Response: We changed "paperboard" to "paper" in Fig. 3 to 9 in response to reviewer comments.

Comment 23.: Fig. 4. SEM micrographs of surface and cross-section of (A1, and A2) paper, (B1, 234 and B2) PBAT-coated paper material. Remark: This Figure shows that the thickness of the uncoated substrate is 140 μm, while the thickness of the coated substrate is 305 μm. Additional remark: Why the thickness of the coated substrate in Figure 4 is 305 μm, and in Table 1 is 419.4 μm? What thickness value is correct?

Response: We performed the changes that were needed as the reviewer suggested. In Table 1 and Fig. 4, the thickness values are corrected.

Fig. 4 shows that the thickness of the uncoated paper substrate is 160 ± 2.0 μm, while the thickness of the PBAT-film coated paper substrate is 439.4 ± 6.0 μm.

Comment 24.: Line 266. The thickness differed from 0.411- and 0.147-mm. Remark: This data is incorrect.

Response: We changed the thickness values of paper and PBAT-film coated paper in according to reviewer advice.

The thickness differed from 0.160 and 0.439 mm.

Comment 25.: The English language also needs to be corrected; Lines 52-53. Polymers have been shown to be ... Remark: “have been to be” should be replaced with “are”.

Response: "have been to be" is changed to "are."

Comment 26.: Line 56. Polymerization ..... produce... Remark: Use here a singular form of the verb “produces” instead of the plural form “produce”.

Response: It has been corrected that "produces" rather than the plural "produce" is used.

Comment 27.: Lines 59-60. According to Rajendran and Han, [19]; and Naser et al., 59 [20], it is therefore believed to be the packaging material which may replace polyethylene... Remark: This sentence must be corrected e.g., as follows, “Rajendran and Han, [19]; and Naser et al., [20], believe that this polymer material can efficiently replace polyethylene”.

Response: We performed the changes that were needed as the reviewer suggested.

Rajendran and Han, [19]; and Naser et al., [20], believe that this polymer material can efficiently replace polyethylene.

Comment 28.: Lines 72-73. ...also extend the shelf-life of food's [28, 29]. Remark: Correct this phrase, as follows, ...” also extending the shelf-life of food”

Response: We accepted the reviewer's suggestions and carried out the changes that were needed.

Comment 29.: Lines 75-76. ...have a less quality than virgin fibers (due to the drying process of the recycled paperboard leads the fiber size... Remark: Replace “less” with “lower” and add “to” before the fiber size....

Response: We carried out the changes that were needed and noted the reviewer's suggestions.

Comment 30.: Line 78. ...via hot pressing... Remark: Replace “via” with “by”.

Response: "via" has been rewritten to "by"

Comment 31.: Line 80. to a 150 °C Remark: Remove the article “a”.

Response: In line 83, the word "a" was deleted.

Comment 32.: Line 81. ... render it... ... Remark: Write “rendering” instead of “render”.

Response: In line 84, "render" has been replaced to "rendering"

Comment 33.: Lines 93-94. ...prior usage... Remark: Add “to” before “usage”.

Response: In line 96-97, "render" has been replaced to "rendering"

Comment 34.: Lime 96. Prior to use, Remark: Replace “Prior to” with “Before”.

Response: In line 96, "prior" has been replaced too "before"

Comment 35.: Line 120. In a N2 atmosphere... Remark: Replace article “a” with “an”

Response: In line 124, "a" has been corrected to "an"

Comment 36.: Line 122. ...used 10.0 mg of material from all of each of the paperboard... Remark: Correct this phrase as follows, ...”10.0 mg of each sample were used” ....

Response: We carried out the changes that were needed and noted the reviewer's advices.

Comment 37.: Lines 128-129. The aim of working with the estimated values was to identify the mechanical property of coated and uncoated paperboard. Remark: This sentence should be corrected as follows, “In this research, the mechanical properties of coated and uncoated paperboard were tested”.

Response: We accepted the reviewer's comments and carried out the changes that were needed.

In this research, the mechanical properties of coated and uncoated paper were tested.

Comment 38.: Line 129. ...characteristics evaluated... Remark: The verb “were” should be added, namely, ...” characteristics were evaluated” ....

Response: The word "were" has been added to line 130.

Comment 39.: Line 130. the TAPPT T494... [38].  Remark:  Write “TAPPI” instead of “TAPPT”.

Response: We made needed corrections; “TAPPI” in line 130

Comment 40.: Lines 130-132. To evaluate the sizes of both coated and uncoated paperboard specimens, 25 × 180 mm of paperboard material had been cut with the paperboard's cellulose fibers. Remark: This sentence should be corrected as follows, “The specimens for the TAPPI T494 test [38] with sizes of 25 × 180 mm were prepared by cutting both coated and uncoated paperboard”.

Response: We performed necessary changes by incorporating the reviewer's comments into consideration.

Comment 41.: Line 136. ...it is also known... Remark: Remove “is”.

Response: In line 136, the word "is" was deleted.

Comment 42.: Lines 136-137. Specimen cutter was used to cut the paper specimens into 100 × 100 mm sizes in accordance with TAPPI T403 [39] standard. Remark: This sentence should be corrected as follows, “To cut specimens into 100 × 100 mm sizes for TAPPI T403 test [39], a special curter was used”.

Response: We implemented the necessary adjustments by considering the reviewer's comments into account.

Comment 43.: Lines 138-139. Bursting strength, which was measured and mentioned in KPa, is the highest pressure reading up to the rupture point. Remark: This sentence should be corrected as follows, “Bursting strength measured in kPa represents the highest strength value achieved before rupture”.

Response: We made needed corrections based on the reviewer's comments.

Comment 44.: 2.3.5. Porosity and Water Absorption (COBB). Remark: Remove (COBB) from this title

Response: The word "COBB" had been removed from the title in Section 2.3.5.

Comment 45.: Additional remark: The text of 2.3.5 contains grammatical errors and should be corrected, e.g., as follows, “The Cobb-60 method was used to measure the water absorption capacity of uncoated and coated paperboard expressed in gH2O per m2 of sample. These measurements were carried out using a COBB tester (Test Techno, India) at room temperature by the TAPPI T456 standard procedure [40]. Samples with the size of 10 x 10 cm were conditioned at 23 ± 1 °C and 50 ± 1% relative humidity. Dividing the Cobb value by grammage yielded the Cobb index.

Response: We made needed corrections based on the reviewer's comments.

Comment 46.: The porosity of the samples was measured using a Frank PTI porosimeter by the TAPPI T 460 standard method. For this purpose, samples with a diameter of 100 mm were prepared. The porosity test of dry samples was carried out at 23 °C under pressure of 1.47 kPa”.

Response: In view of the reviewer's comments, we completed needed adjustments.

Comment 47.: Line 156. method was used... Remark: Replace “was with “were”.

Response: In line 157, "was" has been changed to "were"

Comment 48.: Lines 157-158. The measurements were achieved several times before averaging the results and tests were performed at RT. Remark: This sentence should be corrected as follows, “The measurements were performed several times to calculate an average result”. Remove the final phrase “and tests were performed at RT”.

Response: We made adjustments that were required due to of the reviewer's comments.

The measurements were performed several times to calculate an average result.

Comment 49.: Lines 160-164. Remark: The text of 2.3.7 contains grammatical errors and should be corrected, e.g., as follows, “OCA-20 of Dataphysics Instruments was used to determine the contact angle of uncoated and coated paperboard samples. Water drops of 1 μl volume were used for these measurements. The image of the drops was captured for 5 seconds.

Response: In view of the reviewer's comments, we completed needed adjustments.

Comment 50.: Lines 165-168. Remark: The text of 2.3.8 contains grammatical errors and should be corrected, e.g., as follows, “The received results were analyzed by the variation ANOVA method using the SPSS statistical software package (SPSS Thailand Ltd., Thailand). The value of P was less than 0.05. To identify significant changes Tukey's multiple evaluation test was used”.

Response: The authors agreed with the reviewer comments, and the grammar correction has been done in the revised manuscript.

Comment 51.: The received results were analyzed by the variation ANOVA method using the SPSS statistical software package (SPSS Thailand Ltd., Thailand). A P value was less than 0.05. To identify significant changes Tukey's multiple evaluation test was used.

Response: The authors agreed with the reviewer comments, and the grammar correction has been done in the revised manuscript.

Comment 52.: Lines 172-173.  Applying two layers of PBAT film and confirmed the paperboard surface was without spot, the uniform coatings had been confirmed [43]. Remark: Correct this sentence, as follows, “After applying PBAT films on paperboard surfaces, the coating uniformity was confirmed [43].

Response: We corrected the needed revisions after considering the reviewer's comments.

Comment 54.: Lines 175-178. Since the uniform coating was confirmed, the outcomes recommended that the coating methods may entered into the fibers, which may have served to improve the barrier properties, etc. Remark: Introduce corrections, as follows, “The results showed that the polymer coating may enter the fibers, which can improve the barrier properties of coated paperboard. It is conceivable that the coating with BPAT films contributed to the superior uniformity results compared to those described by Hashmi et al., [45].

Response: The authors agreed with the reviewer's comment. As suggested by the reviewer, the corrections replaced in the revised manuscript.

Comment 55.: Lines 181-183. However, as PBAT is hazardous to chloroform and is difficult to dissolution in less hazardous solvents, with PBAT as a solution is not considered to be a cost-effective. Remark: This sentence contains grammatical errors and should be corrected, as follows, “However, the chloroform is a hazardous solvent and PBAT is difficult to dissolve in it; therefore, using PBAT solution for film forming is not considered cost-effective”.

Response: The reviewer's comment had been accepted by the authors. The modifications have been altered in the revised manuscript on the recommendation of the reviewer.

Comment 56.: Line 192. ...occur nearby 3424 cm−1, ... Remark: Remove “by” and write “... occur near 3424 cm−1, ...”

Response: In line 190, "nearby" has been replaced too "near" occur near 3424 cm1,

Comment 57.: Lines 201-202. ...with respect to the symmetric... Remark: Write “concerning the symmetric” ... instead of ...” with respect to...”

Response: In lines 198–199, the authors agreed with the reviewer's remark. The revised manuscript contains the proposed corrections.

Comment 58.: Lines 212-214. Whereas, the characteristic crystalline peak (2θ = 13.98°) for PBAT seems too weak to be evident in the material used for packaging in the current study due to plastic processing. Remark: Correct this sentence, as follows, “The characteristic crystalline peak of PBAT at 2θ = 13.98° seems too weak to be evident in the material used for packaging in the current study”.

Response: In lines 209–210, the revised manuscript contains the proposed corrections.

Comment 59.: Lines 217-218. Images from SEM can be used to exhibit when the PBAT coating affect the paperboard. Remark: Correct this sentence, as follows, “SEM images can be used to demonstrate the effects of PBAT coating on paperboard morphology”.

Response: The recommended corrections have been incorporated at lines 212–213 in the revised manuscript.

Comment 60.: Line 223...similar results on the smooth surface were obtained [50, 51]. Remark: Replace “on” by “with” as follows, “similar results with the smooth surface were obtained”.

Response: The revised manuscript has added the suggested modifications at line 219.

Comment 61.: Lines 224-225.  Whereas, there wasn't no vacant space in the paperboard's coating surface as the cellulose fiber pores were fully filled with PBAT. Remark: Correct this sentence as follows, “However, there was no vacant space on the surface of the coated cardboard because the pores of the cellulose fibers were filled with PBAT”.

Response: The revised manuscript has added the suggested modifications at lines 221-222.

Comment 62.: Line 245. ... induced the weight loss to increase to 7% to 13%. Remark: Write instead, ... “induced the increase in weight loss to 7- 13%.

Response: In lines 239–240, the authors agreed with the reviewer's comments. The revised manuscript with suggested corrections.

Comment 63.: Lines 245-246. The initial thermal degradation of the paperboard material was observed at 98.4°C due to water that... Remark: Correct this fragment, write “Noticeable weight loss in the paperboard samples was initially observed at 98.4°C due to evaporation of water that...”

Response: The reviewer's opinions are agreed on the authors at lines 240–241. The revised manuscript with recommended corrections.

Comment 64.: Lines 252-258. Remark: This fragment contains many grammatical errors and needs major correction, as follows, “The DSC thermograms of uncoated and coated paperboard samples are presented in Fig. 5(B). When heating the original cardboard, weak endothermic effects were observed, probably due to the water evaporation and thermal decomposition of hemicelluloses. The endothermic peak at a temperature of 170.4oC is related to the melting point of the PBAT film”.

Response: As advised by the reviewers, we performed the needed changes in lines 246-250.

 Comment 65.: All text should be carefully checked and corrected.

Response: The manuscript had been smoothed for grammatical, format, and errors, and the revised version of the manuscript was corrected by a native English speaker.

Round 2

Reviewer 2 Report

Comments and Suggestions for Authors

The manuscript was revised according to the remarks of the reviewer. Therefore it can be recommended for publication.  

Comments on the Quality of English Language

The English errors were corrected.